# Pleiotropic Role of Notch Signaling in Human Skin Diseases

**DOI:** 10.3390/ijms21124214

**Published:** 2020-06-13

**Authors:** Rossella Gratton, Paola Maura Tricarico, Chiara Moltrasio, Ana Sofia Lima Estevão de Oliveira, Lucas Brandão, Angelo Valerio Marzano, Luisa Zupin, Sergio Crovella

**Affiliations:** 1Institute for Maternal and Child Health—IRCCS “Burlo Garofolo”, 34137 Trieste, Italy; rossella.gratton@gmail.com (R.G.); luisa.zupin@burlo.trieste.it (L.Z.); sergio.crovella@burlo.trieste.it (S.C.); 2Department of Medical Surgical and Health Sciences, University of Trieste, 34149 Trieste, Italy; 3Dermatology Unit, Fondazione IRCCS Ca’ Granda Ospedale Maggiore Policlinico, 20122 Milan, Italy; chiara.moltrasio@policlinico.mi.it (C.M.); angelo.marzano@unimi.it (A.V.M.); 4Laboratory of Immunopathology Keizo Asami—LIKA, Federal University of Pernambuco, Recife 50670-901, Brazil; anasofialima2@hotmail.com; 5Department of Pathology, Federal University of Pernambuco, Recife 50670-901, Brazil; lucabrand@gmail.com

**Keywords:** Notch pathway, skin disorder, proliferation, differentiation

## Abstract

Notch signaling orchestrates the regulation of cell proliferation, differentiation, migration and apoptosis of epidermal cells by strictly interacting with other cellular pathways. Any disruption of Notch signaling, either due to direct mutations or to an aberrant regulation of genes involved in the signaling route, might lead to both hyper- or hypo-activation of Notch signaling molecules and of target genes, ultimately inducing the onset of skin diseases. The mechanisms through which Notch contributes to the pathogenesis of skin diseases are multiple and still not fully understood. So far, Notch signaling alterations have been reported for five human skin diseases, suggesting the involvement of Notch in their pathogenesis: Hidradenitis Suppurativa, Dowling Degos Disease, Adams–Oliver Syndrome, Psoriasis and Atopic Dermatitis. In this review, we aim at describing the role of Notch signaling in the skin, particularly focusing on the principal consequences associated with its alterations in these five human skin diseases, in order to reorganize the current knowledge and to identify potential cellular mechanisms in common between these pathologies.

## 1. Introduction

Notch signaling is a ubiquitous and evolutionarily conserved intracellular pathway involved in the regulation of many diverse cellular functions and is implied in mediating responses that might be significantly variable and strictly linked to the activation context and the cellular type [1]. Despite the fact that the Notch pathway operates primarily in the regulation of crucial cellular processes, including proliferation, differentiation, migration, cellular fate and death, both during normal development and pathological conditions, it possesses a relatively simple signal transduction route [2].

Canonical Notch signaling occurs through a cell-to-cell communication in which transmembrane receptors are activated by transmembrane ligands located on the adjacent cell; subsequently, a few molecular components are involved in the transmission of signals from the cellular surface to the transcriptional machinery [3].

Nevertheless, other mechanisms of Notch signaling have been disclosed and are known to occur in a ligand- or transcription-independent manner, generally referred to as non-canonical Notch signaling [4].

To date, a deep characterization of Notch transduction is quite appealing since a progressively expanding spectrum of human diseases has been found to be associated with alterations in Notch signaling [3]. These variations are either due to aberrant regulation or direct mutations leading to both hyper- or hypo-activation of the transcription of target genes or of the core components of the Notch signaling route [3].

In humans, up to 40 different proteins (45 genes) are reported to be involved in Notch signaling [5]. In this review, we will describe the core components, key steps and principal modulators of Notch transduction to then disclose the impact and consequences of their alterations in human skin diseases. In this context, we intend to reorganize the current knowledge on the principle exerted functions of Notch pathway and to identify potential mechanisms in common between these diseases.

### 1.1. Core Components of Notch Signaling

Mammalian cells are known to express four different Notch receptors—Notch1–4, each encoded by a different gene—that together constitute the core of Notch signaling. The four Notch paralogs exhibit both unique and redundant functions and show well-conserved structural homology regions [6]. In general, Notch1–4 receptors are known to exert fundamental roles during cell fate determination by affecting differentiation, proliferation and apoptotic programs. Nevertheless, each receptor is also involved in responses that go beyond cell fate determination and that might depend on their tissue distribution. 

All Notch receptors are single-pass type I integral membrane proteins composed by a series of complex domains and organization motifs [7]. Notch extracellular domain (NECD) comprises the N-terminal extracellular region of the receptors acting as the site of interaction with the ligand located on the signal-sending adjacent cell. NECD is formed by 29 up to 36 epidermal growth factor-like (EGF-like) tandem repeats, sites directly involved in associations with ligands [8] and carries a subset of calcium ions binding sites known to modulate the affinity and conformation of the receptor during ligand binding [9]. Always within the extracellular region, EGF-like tandem repeats are directly followed by the negative regulatory region (NRR), composed by three cysteine-rich Lin12-Notch repeats (LNR-A, -B, -C) and a heterodimerization domain (HD) [6,9]. The NRR is defined as an activation switch for the receptor since it allows a context-specific activation and prevents the induction of a ligand-independent response [10,11]. Receptors possess a single transmembrane domain (TMD) and an intracellular domain (NICD) consisting of a Recombination Signal Binding Protein for Immunoglobulin Kappa J Region (RBP-Jκ) association module (RAM), seven ankyrin repeats (ANK), two nuclear localization signals (NLS) and a transactivation domain (TAD) that contains conserved proline/glutamic acid/serine/threonine-rich motifs (PEST) [9,12].

Following the induction of Notch signaling, the RAM module, together with the ANK motifs and the TAD, is indispensable for the modulation (induction or inhibition) of the appropriate expression pattern of target genes by interacting with transcriptional activators in the nucleus [13]. The NLS motif is essential for the nuclear targeting of the NICD [14], while the PEST domain modulates the proteolytic degradation of this active fragment [13]. 

Differences between the four Notch receptors reside primarily in structural dissimilarities in the NICD. This region is known to affect both the binding affinity between the NECD with its ligand and the interactions of receptors with the transcriptional factors in the nucleus, therefore influencing the cellular and temporal expression of the receptors and of downstream target genes. Structural variations in the NICD imply that the γ-secretase cleaves the intracellular partition of receptors in correspondence of diverse amino acid sequences, therefore generating and releasing from the cellular membrane peptides with distinct amino-termini [15]. The formation of structurally diverse active NICD moieties allows combinatorial interactions with transcription complexes and local/tissue-specific activator proteins bound to regulatory elements of target genes, thus permitting the induction of transcription in a selective manner [15,16]. These specific interactions are known to be essential for the activation of the expression of distinct subsets of target genes; indeed, upon Notch pathway induction, in absence of local/tissue-activators, an insufficient initiation of gene transcription patterns has been reported (Figure 1A) [15,17]. 

Notch signaling in mammals is primarily induced by five functional canonical Notch ligands, all of which are single-pass integral membrane proteins belonging to the Delta-like family of ligands (Delta-like 1 ligand (DLL1), Delta-like 3 ligand (DLL3) and Delta-like 4 ligand (DLL4)) and to the Serrate family of ligands (Jagged1 and Jagged2). These ligands are members of the Delta/Serrate/Lag-2 (DSL) family and are therefore generally referred to as DSL ligands [18].

DSL ligands present a distinctive distribution and expression patterns in the various organs of the adult human body. Different DSL ligands are able to induce differential responses by regulating the levels of the active NICD fragment; this aspect is strictly driven by the ligand identity and the cellular context [19]. Indeed, structural differences between the extracellular domains of DSL ligands define dissimilarities in their signaling strengths, resulting in distinct NICD levels [19]. 

Human DSL ligands have a common structural organization in their extracellular domain comprising an N-terminal domain (NTD), followed by a DSL motif and a variable number of EGF-like repeats ranging from 16 in the Serrate family to 5–9 in the Delta-like family of ligands [20,21]. The various structural motifs found in the extracellular domain are known to be strictly required to guarantee ligand binding to Notch receptors [22].

Specifically, the DSL region is defined as a degenerate EGF-like repeat which is known to be essential but not sufficient to mediate ligand binding to Notch receptors, but it is of crucial importance since mutations in the conserved residues of this domain have been associated with a loss of function in Notch signaling both in invertebrates and vertebrates [23]. 

Another required structural and functional component found in the extracellular region is given by the Delta and OSM-11-like region (DOS) flanking the DSL motif. DOS is composed of the first two EGF-like tandem repeats that are thought to be implied in the activation of the ligand by interacting with the adjacent DSL region [21,22] region.

Furthermore, a relevant structural difference in the extracellular region resides in the presence of a juxtamembrane cysteine-rich domain (CRD) in Serrate ligands which is absent in the Delta-like family members [21].

The C-terminal intracellular region of DSL ligands is poorly conserved and lacks in highly homologous motifs except for numerous lysine residues required primarily for ligand signaling activity, which are found in most but not all DSLs [24]. Regions rich in lysine residues are crucial sites involved in the addition of ubiquitin by the E3 ligase [22]. In addition, Jagged1, DLL1 and DLL4 posses a PSD-95/Dlg/ZO-1 (PDZ) motif that is crucial for interactions with the cytoskeleton (Figure 1B) [24].

Amongst the members of the Delta-like family of ligands, DLL1 and DLL4 present almost the same structural organization, but only about 60% of their protein sequence is the same [25]. The impact of these differences influences primarily their affinity for Notch receptors, which differs more than 10-fold and therefore affects their mutual signaling strengths [26]. Moreover, in the case of Jagged1 and Jagged2 ligands, their constitutive domains and organization motifs are almost the same, though they possess 53% of protein sequence identity. If compared to the other DSL ligands, DLL3 is the most structurally divergent since it possesses a degenerate DSL domain. In addition, DLL3 is predominantly localized in the Golgi apparatus, and it seems to be exposed on the cell surface only when overexpressed primarily under pathologic conditions and acts by inactivating Notch signaling [22,27]. DLL3 inhibitory activity is exerted by maintaining or redirecting Notch receptors and DLL1 in the Golgi apparatus or in the lysosomal and/or late endosomal compartments, thus impeding their insertion in the cell membrane [28].

Non-canonical ligands of Notch receptors have also been identified lately, and they comprise proteins lacking DSL and DOS domains and include secreted proteins as well as integral and glycosylphosphatidylinositol (GPI)-linked membrane proteins. Nevertheless, the physiological functions and molecular mechanisms underlying their activity in the context of non-canonical Notch signaling are not yet fully understood [9].

### 1.2. Notch Signaling

As Notch signaling exerts a fundamental role in many cellular processes and in a vast variety of tissues, it is not surprising that a loss or gain of Notch pathway has been directly linked to many human disorders including developmental syndromes and adult-onset diseases. 

To date, the main steps of a canonical Notch signaling are well established and include a ligand induction of Notch receptor, proteolysis of the NICD, migration of NICD to the nucleus and subsequent activation of downstream target genes. Nevertheless, Notch receptors are able to function non-canonically, in a ligand- or transcription-independent manner, through cellular and molecular mechanisms that are still under investigation and collectively known as non-canonical Notch signaling [29].

The progression of both canonical and non-canonical Notch pathways requires a series of events ranging from the maturation of receptors to their activation. The maturation of Notch receptors occurs through a series of proteolytic cleavages that arise in proximity of the TMD during trafficking to the cell surface. Specifically, by advancing through the secretory route, Notch receptors are cleaved by furin-convertases within the trans-Golgi network at site 1 (S1), generating heterodimers composed by the NECD and a Notch transmembrane and intracellular domain (NTMIC) joined by a non-covalent linkage [30]. Before the final exportation of Notch receptors to the cellular surface, the extracellular domain of the cleaved polypeptide undergoes O-linked and N-linked glycosylations, post-translational modifications known to promote proper protein folding and subsequent interactions with extracellular ligands [31,32].

The activation of Notch signaling following ligand binding causes an irreversible dissociation between the ligand-binding partitions of the receptor from the NICD. As a consequence, each receptor molecule can only signal once; therefore, the advancement of Notch pathway is guided by a strict regulation of Notch proteins in order to obtain a balance between their production and degradation [33].

#### 1.2.1. Canonical Notch Signaling

Canonical Notch signaling is triggered by the binding of Notch receptors to a DSL ligand exposed on a neighboring cell, which leads to a proteolytic processing of the membrane-bound receptor at cleavage sites located in the HD domain, referred to as site 2-4 (S2-4), to ultimately allow the release of an intracellular active fragment, the NICD. 

Following ligand binding, signaling is initiated when the trans-endocytosis of ligand-receptor complexes in the neighboring cell induce a conformational change in the juxtamembrane NRR, ultimately leading to the exposition of S2 and therefore rendering this region susceptible to the proteolytic cleavage catalyzed by a disintegrin and metalloprotease, namely ADAM protease, which removes the extracellular domain bound to the DSL ligand. As a consequence, Notch cleavage at S2 on the receptor-expressing cell generates a membrane-tethered partition of Notch comprising a transmembrane and an intracellular region, namely Notch extracellular truncation (NEXT). The cleavage ultimately leads to the exposure of S3 and S4 sites, which are recognized and cleaved by the γ-secretase intramembrane complex, therefore releasing the NICD [9,34].

Once freed from the membrane, NICD migrates to the nucleus and binds to a conserved DNA-binding protein, CBF1/RBP-Jκ/Su(H)/Lag-1 (CSL or RBP-Jκ in vertebrates), that under basal conditions is known to function as a transcriptional repressor by associating with ubiquitous co-repressor (Co-R) proteins [3,16]. Interactions between NICD and RBP-Jκ are thought to promote conformational changes in RBP-Jκ that facilitate its dissociation from transcriptional repressors and furthermore allow the recognition of transcriptional activators of the Mastermind-like (MAML) family, generating the ternary Notch transcription complex (NTC). The NTC is further involved in the recruitment of general transcription factors to induce the transcription of downstream primary target genes, including the hairy/enhancer of split (*HES*) genes and the hairy/enhancer of split with YRPW motif (*HEY*) genes [9,12,35,36]. HES and HEY proteins are basic helix–loop–helix transcription repressors that act as Notch effectors by negatively regulating the expression of downstream target genes such as tissue-specific transcription factors that affect critical cellular and developmental decisions regarding neurogenesis, blood vessel formation, heart development and somitogenesis [37].

A peculiar feature of Notch signaling resides in the fact that upon ligand binding, the proteolytic cleavage of the NICD induces an irreversible dissociation of the intracellular active signaling unit from the ligand-interacting unit [33]. As a consequence, each receptor can only signal one time, and once released, the NICD fragment can no longer be subjected to regulatory processes mediated by ligand binding or other cell-surface interactions; thus, the turnover of NICD is strictly regulated in order to avoid the presence of a sustained and long-lasting or excessively high signaling [33]. Specifically, the MAML component of the NTC regulates the duration of the transcriptional event by interacting with the cyclin-dependent kinase 8 (CDK-8). The phosphorylation of the PEST domain of NICD fragment by CDK-8 causes the disassembly of the NTC and results in an attenuation of the response. The final step comprises the ubiquitination of NICD by the E3 ubiquitin ligase Fbw7, which leads to its proteasome-dependent degradation [38]. Under physiological conditions, unbound Notch receptors are recycled or degraded through the lysosomal machinery (Figure 2) [38].

#### 1.2.2. Non-Canonical Notch Signaling

Canonical and non-canonical Notch signaling need to be considered as two strictly integrated pathways triggered by a single receptor [39]. Indeed, the precise interplay between the principle activities exerted by canonical Notch signaling during cell fate determination and non-canonical Notch signaling, which primarily include cell adhesion, cytoskeletal remodeling and cell motility, needs to be taken into account to unravel the complex functions of Notch signaling cascade [39,40].

Nevertheless, while the molecular mechanisms and cellular functions of a canonical Notch pathway have been well established, the signaling events underlying a non-canonical Notch cascade have yet to be fully elucidated. 

Recent studies highlighted the necessity to undertake a thorough characterization of the non-canonical mechanisms of the Notch pathway since its alterations have been seen to be potentially associated with different pathological conditions including cancer and immune deregulations [4,41]. Indeed, it might be possible that a deep understanding of the non-canonical Notch signaling could be useful to assess specific strategies able to block pathological Notch signaling while simultaneously maintaining intact many other physiological processes mediated by the canonical Notch route [4]. 

The better-characterized modes of non-canonical Notch signaling include regulated activation of Notch route by the γ-secretase occurring independently from ligand interaction, RBP-Jκ-independent activity of NICD, induction of Notch signaling induced by a membrane-bound form of Notch receptor in absence of a proteolytic cleavage mediated by the γ-secretase protease or in some cases of ligand interaction [4]. 

The ligand-independent activation of Notch receptors is known to occur through endosomal trafficking. The levels of Notch proteins are strictly regulated in order to obtain a balance between production and degradation. In the absence of an interaction with a ligand, the NICD is marked for internalization and degradation by ubiquitination mediated by the E3 ubiquitin ligase Deltex [42]. Indeed, the turnover of Notch receptors under physiological conditions requires that the ubiquitinated Notch proteins are sorted from the endosomal vesicles into the intraluminal vesicles of the multivesicular bodies (MVBs). Once in the lumen of the MVBs, Notch receptors are transferred into the lumen of lysosomes for subsequent degradation. Nevertheless, it might occur that while being directed towards the lysosomal compartments, a disturbance of the endosome-mediated sorting of Notch receptors might result in the triggering of a ligand-independent activation of the receptor [43]. In this context, Notch receptors are retained in the endosomal membrane site, in which they may be subjected to an accidental intracellular activation following the removal of the NICD, which mimics the modes of a ligand-dependent Notch induction [42,43]. 

Notch signaling might also occur through an RBP-Jκ-independent mechanism that seems to be crucial in cellular processes including primarily immune dysfunctions and oncogenesis [4]. In this context, the released NICD active fragment might regulate transcription principally through two alternative mechanisms: the first contemplates the interaction of the active NICD with transcription factors not belonging to the CSL/RBP-Jκ family, including hypoxia-inducible factor (HIF) and monocyte enhancer factor-2 (Mef2) [44]; the second involves the delivery into the cytoplasm of a slightly different fragment from NICD resulting from the cleavage catalyzed by a distinct protease from presenilin of the γ-secretase complex [41,44]. 

The protease-independent activation of Notch signaling occurs when Notch receptors are not processed by the γ-secretase complex [41]. It is widely accepted that the cleavage of Notch receptor by the catalytic moiety of the γ-secretase constitutes a hallmark of canonical Notch activation. Nevertheless, different studies, showing limited evidence so far, suggest the presence of a non-canonical Notch signaling that is activated in absence of the cleavage catalyzed by the γ-secretase complex, revealing that, in limited specific occasions, Notch can be non-canonically processed [41]. 

## 2. Skin

The skin is the most extended human organ, and it is partitioned into two layers: the epidermis, the uppermost layer constituted by a stratified squamous epithelium primarily composed of keratinocytes and dendritic cells; and the derma, the bottom layer composed of connective tissue bearing vascular and nervous networks, resident fibroblasts, mast cells and macrophages, amorphic collagen and epidermal appendages [45]. 

Specifically, the epidermis is defined as a multi-layered epithelium, comprising the interfollicular epidermis (IFE) and associated epidermal appendages (hair follicles, sebaceous and sweat glands). The IFE is primarily composed of progressively differentiated keratinocytes organized in specific layers that, from the deepest to the most superficial one, are given by the basal layer, spinous layer, granular layer and the stratum corneum (Figure 3) [46]. 

The epidermal stem cells (ESC) reside in the basal layer and are represented by mitotically active cells that yield at every division cycle either more stem cells that self-renew or transient amplifying cells, defined as a cellular progeny that undergo terminal differentiation strictly driven and regulated by local microenvironmental signals, to ultimately generate IFE, hair follicles and apocrine and sebaceous glands [46]. During differentiation, the transient amplifying cells stop proliferation and progressively migrate upwards to the stratum corneum. During this migration, the keratinocytes acquire layer-specific characteristics, including the expression of peculiar proteins such as specialized epidermal keratins like keratin 1 (K1), keratin 5 (K5) and keratin 10 (K10); transcriptional activators including NF-κB and peroxisome-proliferator activated receptor (PPARγ); involucrin (IVL); transglutaminase (TGM) 1; periplakin (PPL), and loricrin [47,48,49,50]. Subsequently, once in the stratum corneum, cells cease their metabolic activity, and as stratification occurs, epidermal appendages such as hair follicles and associated sebaceous or sweat glands are formed [51,52].

### 2.1. Notch Signaling and Skin

Notch activation regulates skin homeostasis by balancing primarily growth arrest and progressive differentiation processes of keratinocytes; indeed, Notch signaling is known to act as a molecular switch that strictly modulates the advancement of cells through the various skin layers during epidermal differentiation [53]. Recent studies highlighted the presence of a specific crosstalk activity between the different Notch receptors, ligands and downstream signaling molecules, including p21, involved in inducing growth arrest of keratinocytes, in the initiation of terminal differentiation and requiring Notch for the induction of its expression [54]; and p63, which acts by promoting keratinocyte differentiation and whose activity is counteracted by Notch to maintain immature cell characteristics [55].

### 2.2. Notch and Skin

Throughout time, the Notch pathway in the skin has been proven to be essential for skin homeostasis [56]. Indeed, Notch receptors and ligands could influence the maintenance of epidermal homeostasis, by providing the correct control of proliferation and differentiation programs within the epidermal cells, during adult and embryonic development [48,52,57]. 

In detail, Blanpain et al. presented molecular and functional in vivo evidence of overall skin homeostasis mechanisms [58]. Their work highlighted the pleiotropic role of Notch signaling in the proliferation and differentiation of the epidermal cells by observing that Notch operates to govern the balance between proliferative basal progenitors and their terminally differentiating progeny, culminating in epidermal barrier formation [58]. 

When ligands bind to Notch receptors, NICD is released and translocates to the nucleus, where it induces the activation of downstream target genes including HES and HEY family of genes, p21 gene inducing keratinocytes’ growth arrest and p63 expression that acts by promoting keratinocyte differentiation [9,12,35,36,59]. Once activated, target genes mediate the regulation of signaling pathways aimed at committing the ESC of the basal layer towards two distinct fates: proliferation with self-renew purposes or differentiation [60,61,62]. 

Notch1–4 receptors and their ligands play an important role in regulating epidermal proliferation and differentiation, and their distinctive expression in the epidermal layers seems to correlate with the activation of layer-specific target genes in a well-determined and distinct manner, therefore strongly impacting the subsequent IFE development [50,57]. 

Notch1, Notch2, Notch3 and Notch4 receptors have been well documented to be expressed in the IFE, predominantly in the suprabasal cells, and the expression of receptors has been seen to occur in proliferating cells or cells that are initiating or undergoing terminal differentiation [59]. Notch ligands detected in the epidermis comprise Jagged1, Jagged2 and DLL1 [59]. Jagged1 expression has been predominantly identified in the suprabasal layer, while Jagged2 and DLL1 ligands have been principally identified in the basal layer (Figure 4) [50,57]. 

Jagged1 and Jagged2 ligands bind to Notch family members with similar specificities and act by promoting cellular terminal differentiation [50].

A differential outcome is given by Notch interactions with DLL1 ligand. DLL1 have been reported to be exclusively expressed in the basal layer, where interactions with Notch receptors results in the maintenance of ESCs in an undifferentiated state [58]. Nevertheless, it is known that upon receptor-ligand binding, DLL1 functions also by blocking Notch signaling [57,58,62]. Notably, the role of DLL1 in the modulation of Notch activity is extremely intriguing. DLL1 counteracts the activity of basal-layer-localized Notch1 receptors by upregulating the cell cycle regulator p21, an inhibitor of cell cycle progression, therefore leading to the downregulation of ESCs proliferation [56,63]. Therefore, keratinocytes are committed to exit from the stem cell compartment, to their subsequent detachment from the basal layer and to initial differentiation [58].

Further agreeing with this idea, Negri et al. noted an incremented proliferation of the ESCs in knockdowns for Notch1 [57]. Rangarajan et al. also detected epidermal hyperproliferation and reduced differentiation in a mouse model with conditional ablation of Notch1 in skin epithelium [56].

Notch1 is known to activate caspase 3, to minimize stem cell proliferation and to increase cellular differentiation [64,65]. In addition, the receptor negatively regulates p63, a p53 family member, in order to prevent an inhibition of differentiation [66]. It is important to note that, also in this case, p63 is expressed in cells of the basal layer where proliferation occurs, while it is strongly down-modulated in the upper layers where differentiation happens [66]. Remarkably, proliferation or differentiation of keratinocytes may be connected to a delicate balance between Notch1 and p63 levels [67].

In contrast to Notch1’s well-reported activity, in several studies using mice carrying conditional mutations in Notch2, -3 and -4, it was observed that the ablation of these 3 molecules does not cause any overt phenotype in the skin [68,69,70].

It is interesting to note that mice deficient in Notch1 or Notch2 gene are embryonic lethal [71,72], while mice deficient in Notch3 or Notch4 are born without any apparent altered phenotype in the skin [68,69]. 

Mazur et al. detected a reduction of p21 expression in a mouse model with ablation of Notch1 that is not found in a model with ablation of Notch2; in fact, they speculated that Notch2 signaling might not be required for skin differentiation since it is expressed exclusively in suprabasal keratinocytes, mainly represented by already differentiated cells [65]. 

Except for the cellular localization of Notch3 and Notch4, no information is available regarding their involvement in skin physiology.

Furthermore, excluding differentiation and proliferation, Notch signaling also regulates other important outcomes in the epidermis such as wound healing through the regulation of vascular endothelial cell proliferation, tube formation and migration of keratinocytes and fibroblasts [73,74].

### 2.3. The Main Proteins Related to Notch and Skin 

Other molecules play a key role in the correct assembly and functioning of Notch signaling. The activity of these molecules is essential for the regulation of many diverse cellular functions. Indeed, their alterations can induce signal failure and, as a consequence, health disorders.

The principal molecules related to Notch signaling in the skin are the following: 

γ-secretase: The maturation and activation of Notch receptors is strictly regulated by a series of proteolytic processings. The γ-secretase complex is a transmembrane protease that exerts a fundamental role during Notch activation by catalyzing the ultimate cleavage of the receptor, causing the release of the intracellular and active fragment NICD, that once migrated in the nucleus, induces the expression of genes involved in epidermal and follicular differentiation and proliferation [75]. The γ-secretase consists of four protein subunits, comprising: presenilins (PSEN), the catalytic subunit; and three cofactor subunits, represented by presenilin enhancer-2 (PSENEN), nicastrin (NCSTN) and anterior pharynx defective-1 (APH1) [76]. These proteins are encoded by six genes: *PSEN1*/*PSEN2*, encoding for PSEN1 and PSEN2; *NCSTN*, encoding for NCSTN; *APH1A*/*APH1B*, encoding for APH1; and *PSENEN*, encoding PSENEN [77]. Mutations in the genes encoding for the different subunits of the γ-secretase complex have been linked to various skin disorders and primarily act by affecting Notch signaling [77]. 

GDP-fucose protein O-fucosyltransferase 1 and Protein O-glucosyltransferase 1: Glycosylation of Notch ECD, by the addition of O-glycans, has progressively emerged as a fundamental post-transcriptional modification able to strictly regulate the receptors’ activity [78]. In the context of canonical Notch signaling, a pivotal role in ECD glycosylation is given by two enzymes: the first is the GDP-fucose protein O-fucosyltransferase 1 (POFUT1), encoded by *POFUT1* gene, a protein involved in catalyzing the addition of an O-linked fucose moiety [79] to the EGF-like repeats of the ECD domain; the second is the protein O-glucosyltransferase 1 (POGLUT1), encoded by *POGLUT1* gene, that functions by adding an O-linked glucose [80] to the EGF-like repeats of the ECD domain. These post-transcriptional modifications allow the binding between Notch ligands and receptors, which ultimately leads to the proteolytic release of NICD. Though the localization of POFUT1 is not uniform in the epidermis, recent evidence strongly suggests that this enzyme may influence melanin synthesis and transport in melanocytes [81]. A high expression of POGLUT1 has been registered in the matured layers of the epidermis, and this aspect may indicate the involvement of POGLUT1 in the differentiation and the development of the epidermis [82]. 

EGF domain-specific O-linked N-acetylglucosamine transferase: The epidermal growth factor domain-specific O-linked N-acetylglucosamine transferase (EOGT), encoded by *EOGT* gene, is an O-linked N-acetylglucosamine (O-GlcNAc) transferase known to transfer O-GlcNAc moieties to consensus sequences in EGF-like repeats of few membrane-bound and secreted proteins, including Notch receptors [83,84]. EGF-specific O-GlcNAc glycosylation is a rare form of functional post-transcriptional modification occurring exclusively in the lumen of the endoplasmic reticulum, and it seems to be essential for Notch regulation and ligand-induced Notch signaling [83]. Pathogenic variants on *EOGT* impair the glycosyltransferase activity of the enzyme, resulting in a defective post-translational modification of Notch receptors [83]. Nevertheless, the impact of the activity of EOGT on Notch receptor and signaling has still to be fully understood, and currently, no specific data are available regarding its involvement and activity in skin physiology. 

Filaggrin: Filaggrin (FILA), encoded by the *FLG* gene, is a late epidermal differentiation protein that plays an important role in the skin’s barrier function. FILA interacts exclusively with intermediate filaments and is specifically known to possess a strong keratin-binding activity; indeed, once FILA binds to intermediate keratin filaments, it causes their dense aggregation into microfibrils, ultimately rendering intermediate filaments tightly packed in parallel arrays. The resulting crosslink between keratin intermediate filaments leads to the formation of highly insoluble keratin, which acts as a protein scaffold for the subsequent attachment of lipids and proteins, necessary to guarantee the progressive differentiation of keratinocytes [85]. In the IFE, terminally differentiated keratinocytes express keratin-bundling protein FILA, in the upper granular layer and in the cornified envelope [85,86]. Notch signaling is required for the late-stage granular layer differentiation and correct filaggrin processing in the epidermis [48]. 

## 3. Notch Signaling and Skin Diseases

Any disruption of canonical and non-canonical Notch cascade, resulting in a gain or loss of function, can induce a health disorder as a consequence of signal failure, as Notch signaling orchestrates the regulation of cell proliferation, differentiation, migration and apoptosis.

The molecular mechanisms through which Notch contributes to the pathogenesis of skin diseases are multiple and are still far from being fully understood. Despite the initial efforts focused on skin cancer, attention has recently turned also on the correlation between Notch signaling and skin diseases other than malignancies.

To date, the skin diseases correlated to alterations in Notch signaling are: Hidradenitis Suppurativa, Dowling Degos Disease, Adams–Oliver Syndrome, Psoriasis and Atopic Dermatitis.

### 3.1. Hidradenitis Suppurativa (HS)

Hidradenitis Suppurativa (HS) is a chronic inflammatory skin disease affecting the hair follicle. 

It has been estimated that the pathology presents a prevalence of 0.05–4% in Europe with a female predominance [87] and with an onset after puberty. Interestingly, in familial cases of HS, the first manifestations can occur in younger children, who often tend to develop a more severe form of the disease [88].

The events underlining the early onset of HS phenotype are given by infundibular hyperkeratosis of the terminal hair follicles, perifolliculitis and hyperplasia of follicular epithelium, events that collectively anticipate the follicular occlusion and disruption. The rupture of hair follicles releases cellular debris and keratins into the surrounding dermis, therefore activating an inflammatory immune response [89,90].

The primary visible lesions of HS are recurrent, painful, subcutaneous and inflamed nodules that can rupture, leading to deep and purulent dermal abscess [91]. With the progression of the disease, the connection of the lesions can result in dilated sinus tracts (i.e., skin tunnels) [92], fibrosis and scarring [91]. HS clinical manifestations occur primarily in apocrine-gland bearing regions at inverse body sites including the axilla, genito-femoral area, perineum, perianal and gluteal region and are characterized by recurrence and chronicity that significantly impact the patients’ quality of life [91].

The exact etiology of HS is yet not completely unraveled; however, both genetic and environmental factors are known to trigger the development of the disease. A recent article by Tricarico et al. reported an exhaustive overview of the genes involved in HS susceptibility [93], highlighting that about 35% of HS patients present a family history of HS. Mutations in *NCSTN*, *PSENEN* and *PSEN1* genes, respectively encoding for NCSTN, PSENEN and PSEN1, have been identified as the most common genetic variants involved in HS familial cases. These three proteins are essential components of the γ-secretase multiprotein complex [94], and their haploinsufficiency is probably linked to nonsense-mediated decay of their mRNA, resulting in a dysfunction of γ-secretase activity and ultimately in alterations of Notch signaling (Figure 5) [94]. 

In an in vitro model of HS, developed through *NCSTN* silencing in HaCaT cells, Xiao et al. observed that *NCSTN* inhibition induced cell proliferation and cell cycle progression, probably through a modulation of phosphoinositide 3-kinase (PI3K)/AKT pathway. The transcriptome profile of these *NCSTN*-silencing HaCaT cells underlined the expression of different genes related to biological processes such as epidermis development, epidermal cell differentiation and keratinocyte differentiation and keratinization. Moreover, a downregulation of genes involved in the Notch signaling pathway was also detected. Similarly, in epidermidal biopsies derived from an HS patient with *NCSTN* mutations, characterized by psoriasiform hyperplasia of the interfollicular epidermis, Notch pathway molecules, such as Notch1-3 and HES-1, were found to be decreased in lesional regions with respect to normal areas [95].

Studies on animal models gave some interesting results regarding the role of Notch pathway in the epidermal environment. For instance, sebaceous gland development was blocked in mice with γ-secretase deficiency [70], whereas hair follicles were replaced by epidermal cysts in mice with Notch deficiency [96].

### 3.2. Dowling Degos Disease (DDD)

Dowling Degos Disease (DDD) is a rare autosomal dominant skin genodermatosis [97].

DDD is very rare, since to date few cases have been reported in the literature, and it possesses a post-pubertal age of onset from the third to fourth decade of life [98].

DDD is characterized by reticulate hyperpigmentation, by small hyperkeratotic dark-brown papules and lentigo-like brown macules [99]. The hyperpigmentation tends to increase progressively over time from flexural sites to intergluteal and inframammary areas, neck, trunk, arms and thighs [99]. The hyperpigmentation occurs in the epidermal basal layer with the thinning of the suprapapillary epithelium. Moreover, melanophages and infiltrates of lymphocytes and histiocytes are present [100], together with melanosomes and keratinocytes with irregular appearance [82].

Classical DDD is caused by mutations in the *KRT5* gene, encoding for keratin 5 (K5) that is normally paired with keratin 14 (K14) to assemble intermediate filaments; in DDD, K5 is truncated, resulting in abnormal intracellular perinuclear architecture of intermediate filaments [99]. Actually, in a breast cancer cell line (MCF7) transfected with plasmid carrying DDD mutation in *KRT5*, K5 was not embodied in cytoskeletal intermediate filament network persisting in its soluble form [99]. Normally, this fraction is small, but it is essential during differentiation for dynamic cytoskeleton rearranging [101]. Therefore *KRT5* haploinsufficiency could negatively impact cell adhesion, organelle movement, nuclear anchoring and melanosome transport into keratinocytes, triggering epithelial remodeling [99]. K5 is linked to Notch pathway; indeed, a loss of K5 expression during epidermal differentiation is concurrent with an increased activation of Notch1 [102]. 

Other genes involved in DDD are *POFUT1*, *POGLUT1* and *PSENEN* [82,103]. Both POFUT1 and POGLUT1 are components of the canonical Notch signaling pathway.

POFUT1 performs O-fucosylation of Notch receptors driving ligand-receptor binding and the proteolytic release of NICD; in turn, NICD, complexing with RBP-Jκ, impacts the expression levels of NOTCH1, NOTCH2 and HES1 [81,104]. Indeed, in shRNA *POFUT1* HaCaT cells, the gene expression of these three genes was downregulated [81]. *POFUT1* mutations in DDD lead to nonsense-mediated decay of the transcript and impaired Notch signaling with production of abnormal skin pigments [81]. 

POGLUT1 catalyzes the O-glycosylation of Notch receptors, therefore impacting the receptors’ conformation and allowing the activation of intracellular pathways [80]. DDD patients carrying heterozygous nonsense or splice site *POGLUT1* mutations probably present a nonsense-mediated mRNA decay resulting in haploinsufficiency. The immunohistochemistry showed a weak expression of POGLUT1 in the epidermal layers in comparison to healthy donors, where the staining is more prominent, especially in the stratum spinosum and granulosum, indicating the role of this protein for the correct differentiation of the epidermidis [82]. Interestingly, in DDD biopsies, an increment of K5 was also observed if compared to healthy controls, possibly suggesting an interplay between Notch signaling and K5 during epidermal maturation [82].

Individuals carrying *PSENEN* mutations present a different phenotype if compared to classical DDD patients. Indeed, genetic variations in *PSENEN* are pathogenetic for the development of HS. Therefore, co-manifestations of DDD and HS were reported for these patients that develop reticulate pigmentation, comedones, follicular hyperkeratosis, nodules and scars [105]. In our recent study, we also described a patient with familial HS and concomitant DDD harboring a novel nonsense mutation in *NCSTN* gene associated with a reduced quantity of subunits of γ-secretase [106]. 

PSENEN is a cofactor of gamma-secretase that is involved in the canonical Notch pathway, by cleaving Notch intracellular domain and activating it [107]. In keratinocytes derived from patients carrying *PSENEN* mutations, the expression of *PSENEN*, *POGLUT1* and other Notch related genes were decreased and associated with an abnormal Notch signaling (Figure 6) [107].

### 3.3. Adams–Oliver Syndrome (AOS)

Adams–Oliver Syndrome (AOS) is a rare inherited disorder, with an estimated incidence of 1 in 225,000 live births, characterized by the combination of aplasia cutis congenita (ACC) of the scalp vertex and terminal transverse limb defects (TTLD) including hypoplastic nails, brachy/oligodactyly and amputation defects [108].

Additional major features of AOS comprehend vascular anomalies such as cutis marmorata telangiectatica congenita (CTMC), pulmonary and portal hypertension and retinal hypervascularization. 

Autosomal dominant or sporadic forms of AOS are linked to mutations in *ARHGAP31* (Rho GTPase Activating Protein 31), *DLL4*, *Notch1* or *RBP-Jκ* genes, while mutations found in *DOCK6* (Dedicator of Cytokinesis Protein 6) or *EOGT* genes characterize an autosomal recessive inheritance of the syndrome [109,110].

Each of these genes is involved in tightly regulated processes occurring during embryonic development that specifically involve Notch signaling pathway and the organization of actin cytoskeleton. Mutations and/or alterations in any one of them can impair these mechanisms, therefore leading to the onset of the AOS phenotype [110].

Notch pathway seems to play a determinant role in the pathogenesis of AOS; indeed, 71% of identified mutations found in cases affect genes directly correlated to Notch signaling, including *Notch1*, *DLL4*, *RBP-Jκ* and *EOGT*, primarily through haploinsufficiency or loss of function mechanisms, by impacting the maturation of receptors, receptor-ligand binding and the transcription of target genes [110]. 

The activation of Notch receptors ultimately leads to the migration of the NICD active fragment in the nucleus and immediately interacts and activates the transcriptional activator RBP-Jκ. Once activated, RBP-Jκ functions by recruiting chromatin remodeling complexes containing histone deacetylase/acetylase proteins to promote the expression of downstream target genes [111]. 

EOGT is an enzyme that catalyzes the addition of O-GlcNAc moieties to consensus sequences in the EGF-like repeats of the ECD domain of Notch receptors during their maturation occurring through the secretory pathway. EOGT has been seen to be particularly relevant for the glycosylation of Notch1 receptors in mammalian cells [112]. EOGT pathogenetic variants cause impaired glycosyltransferase activity, resulting in a defective post-translational modification of targeted Notch proteins directed towards the plasma membrane [113]. However, the impact of these variants on Notch signaling remains still poorly understood.

*Notch1* mutations are frequent and likely constitute the primary single genetic origin of the disease [114]. *Notch1* can present deleterious genetic variants, including deletions, frameshifts, missense, nonsense and splice site mutations. Truncating mutations, which are generally spread throughout the entire length of the gene, result in the degradation of the mutant transcript through the nonsense-mediated decay mechanism. Clusters of missense mutations occurring in the regions encoding for EGF-like tandem repeats 11–13 have been identified, and they have been seen to impair the binding of Notch1 to DSL ligands [115]. Furthermore, missense variants often cause the addition or removal of cysteine residues, and the presence of an abnormal amount of this amino acid causes an alteration in the formation of disulfide bonds, therefore disrupting the tertiary structure of the receptor [115].

DLL4 is a critical Notch ligand, encoded by the *DLL4* gene, that is capable of binding and promoting the activation of Notch1 and Notch4 receptors. Recent studies identified heterozygous pathogenic variants and nonsense and missense mutations in *DLL4*. Specifically, most of these missense variants have been characterized and seen to cause a replacement or creation of cysteine residues, and as a consequence negatively impact the structural integrity of the ligand [108]. These findings strongly suggest that sequence changes in this gene are responsible for the onset of AOS [108]. Another relevant function of this gene comprises its involvement in the negative regulation of endothelial cell proliferation, angiogenic sprouting and retinal progenitor proliferation [116]. 

Mutations in *RBP-Jκ* are commonly given by missense variants that affect the DNA-binding region of the transcriptional activator, which is no longer able to bind to the promoters of target genes, specifically of *HES1*. The resulting impaired binding ability of RBP-Jκ leads to a disruption of the transcriptional regulation of Notch signaling on downstream target genes [117].

On the contrary, the remaining 29% of patients carry causative variants in *ARHGAP31*, encoding for the Rho GTPase-activating protein 31 (ARHGAP31), and *DOCK6*, encoding for guanine nucleotide exchange factor (GEF), which are not directly associated to Notch signaling. These genes encode for regulatory proteins that are actively involved in actin cytoskeleton formation and therefore exert a primary role in cell morphology, cellular migration, cell division and survival [110,118]. Despite six genes underlying AOS having been well-established and characterised to date, recently it has been observed that some probands do not possess mutations in any of these genes (Figure 7) [110].

### 3.4. Psoriasis

Psoriasis is a chronic immune-mediated, proliferative and inflammatory disorder with primary cutaneous manifestations and a strong genetic predisposition. This skin disorder is characterized by papules and plaques in demarcated areas of affected skin with variable morphology, distribution and severity. The scalp, elbows, knees, lower back, hands, feet and body folds are commonly affected sites, and the lesions have typically a symmetrical distribution [119].

The pathogenesis of Psoriasis is complex and multifactorial and is the result of an interplay between disturbances in the innate and adaptive cutaneous immune responses, complex genetic and epigenetic backgrounds and alterations in the skin microbiome [120]. 

The major events underlying Psoriasis’ etiology comprise an abnormal differentiation and proliferation of keratinocytes causing epidermal hyperplasia, dermal infiltration by various immune cells, and increased dermal capillary density leading to an augmented permeability in wide-caliber vessels [121,122,123]. 

To date, the major accredited immunopathogenic mechanism of Psoriasis asserts that the crosstalk between keratinocytes and autoreactive T cells leads to the development of inflammatory and immune-driven responses that are necessary for the onset, progression and persistence of the disease [124]. The ongoing infiltration in the epidermis and dermis of different leukocyte populations causes hyperproliferation of the epidermis and an altered keratinization, which results in a thickened epidermis, altered constitution of the cornified envelope of the skin and elongated protrusions in the dermis [124,125]. Nevertheless, the sole T cell-induced immune response targeting keratinocytes cannot justify the phenotype of Psoriasis. Indeed, growing evidence suggests that intrinsic alterations in epidermal keratinocytes also have a relevant impact in the development of the disease [126]. The intrinsic alterations found in keratinocytes of Psoriasis cases have been seen to affect primarily the expression of cytokines and growth factors that directly convert T cell-derived signals in a dysregulated hyperproliferation and differentiation; induce the activation, recruitment and retention of T cells in the epidermal compartment; and promote angiogenesis [124,126,127].

Bearing in mind that the proliferation of ESCs and the progressive differentiation of keratinocytes are strictly regulated by the Notch signaling pathway, it is not surprising that alterations in this intracellular route have been found to be crucial in the onset of Psoriasis [128].

Further sustaining the potential crucial role of Notch cascade in the pathogenesis of this disorder is a study conducted by Thélu et al. [128], in which authors registered a decrement in Delta-like/Notch signaling in conditions of altered cell proliferation and differentiation in Psoriasis skin lesions and in in vivo experiments that involved the grafting of normal human skin on a nude mouse [128].

Ota et al. [129] observed a decrement in Notch molecules in psoriatic skin that might cause an aberrant expression and localization of K10 and K14, leading to abnormal differentiation of the epidermis. Moreover, they observed a decrease in Notch1 and Notch2 expression that caused hyperproliferation of keratinocytes through the induction of *p21* gene expression. Psoriatic skin is characterized by hyperproliferation and aberrant differentiation; in fact, these events comprise the typical histopathological basis for the formation of epidermal hypertrophy and parakeratosis [130].

Instead, Skarmoutsou et al. [131] have registered a hyperactivation of the Notch pathway in skin samples derived from psoriatic patients. In particular, Skarmoutsou et al. showed a significantly higher protein expression level of Notch1, Notch2, Jagged1 and hairy/enhancer of split 1 (Hes1) in psoriatic skin lesions if compared to normal controls. Abdou et al. [132] also recently observed an upregulation and not a downregulation of Notch1 in 35 lesional biopsies of psoriatic patients in comparison with normal skin biopsies. In addition, in a study conducted by Jiao et al. [133], authors found a nuclear form of Notch1 that guarantees the activation of the cascade of Notch signaling and whose levels are significantly associated with the severity of Psoriasis diseases. Furthermore, in another recent work conducted by Rooney et al. [134], the role of Notch1 signaling was investigated in the context of Psoriasis by studying the tissue expression of several effectors of this route such as Notch1, DLL4, Jagged1, vascular endothelial growth factor (VEGF) and other proteins. The results of this study highlighted an increased expression of Notch1, DLL4, Hrt-1 (Ring-box protein HRT1) and A-SAA (Acute-phase Serum Amyloid A) in Psoriasis lesional skin if compared to unaffected skin. Moreover, the authors observed that A-SAA plays a role in the transcriptional regulation of *Notch1*, and it induces angiogenesis and vascular invasion, a process negatively regulated by Notch1 siRNA. The authors concluded that vascular dysfunction in Psoriasis is mediated by Notch1 signaling route and that this pathway could represent a novel therapeutic target (Figure 8). 

### 3.5. Atopic Dermatitis (AD)

Atopic Dermatitis (AD), also called Atopic Eczema, is one of the most common chronic pruritic inflammatory diseases and affects up to 20% of children and adolescents worldwide and oscillates between 2% and 17% in the adult population [135].

The clinical presentation of AD varies widely; indeed, AD patients may exhibit various symptoms including pruritus, xerosis, pain and sleep disturbance, which lead to a severe impairment in quality of life. Furthermore, the progression of the disease is chronic but intermittent [136].

AD is associated with comorbidities such as asthma, allergic rhinitis, food allergies and an increased risk of other inflammatory diseases, such as arthritis and inflammatory bowel disease, though to date the link between these diseases is not yet known [137].

The pathogenesis of AD is multifactorial, and it is thought to occur via a combination of skin barrier abnormalities; immune dysregulation, including excessive T helper-2 cell activity; and genetic and environmental factors [138,139].

A family history of AD is the strongest known risk factor for the development of the pathology [140]. Mutations in the *FLG* gene have been identified as the most common genetic variations involved in AD familial cases (20–30% of AD patients compared with 8–10% of the general population without AD) [86]. *FLG* encodes for a large protein called profilaggrin that is cleaved to produce multiple copies of the functional FILA protein, which plays an important role in the skin’s barrier function [86]. Loss-of-function mutations in the *FLG* gene lead to the truncation of profilaggrin and to a loss of FILA expression.

Major insights into AD reveal an important role for disturbed epidermal differentiation with impaired skin barrier function in the pathophysiology of the disease [141]. Interesting evidence shows that Notch signaling is required for late-stage granular layer differentiation and correct FILA processing in the epidermis [48]. Confirming the close correlation between FILA and Notch signaling, Dumortier et al. observed a downregulation of all Notch receptors expression in the epidermis of lesional skin of AD, whereas healthy control patients exhibited significant Notch expression confined to suprabasal epidermal layers [142]. The same authors confirmed the main role of Notch signaling in AD with skin-specific simultaneous inactivation of Notch1 and Notch2 that induced the development of an AD-like disease in a mouse model. In fact, these adult mice were characterized by acanthosis, dry skin, spongiosis, hyperkeratosis and massive dermal infiltration of eosinophils and mast cells (Figure 9) [142].

## 4. Conclusions

With the collaboration of other cellular pathways, Notch signaling orchestrates the regulation of cell proliferation, differentiation, migration and apoptosis of the epidermal cells. Any disruption of canonical and non-canonical Notch signaling, resulting in an up- or downregulation of Notch, either due to aberrant regulation or direct mutations that might lead to both hyper- or hypo-activation, can induce skin diseases. Indeed, the role of Notch signaling can be considered as pleiotropic in light of its involvement in proliferation as well as in differentiation of epidermal cells. In the previously described skin diseases, the relationship between Notch activity, keratinocyte proliferation and differentiation has been thoroughly described, and they gave rise to contradictions: Notch has been reported to be downregulated in all diseases discussed herein, with the exception of Psoriasis and DDD, in which either an upregulation or a downregulation of Notch signaling has been reported. The identified down- or upregulation induced hyperproliferation of keratinocytes in HS, Psoriasis, AOS and AD and abnormal differentiation of keratinocytes in HS, DDD, Psoriasis and AOS; instead, in AD there is a disturbed differentiation in association with decreased epidermal barrier function (Figure 10).

Accumulating evidence suggests that alterations of Notch signaling play a crucial role in the pathological features of these skin disorders. The understanding of the exact roles exerted by Notch signaling in all the single skin diseases is important to characterize potential common mechanisms between them, in order to better correlate cell signaling dysfunctions with clinical features and to contribute to the identification of targeted pharmacological intervention.

Unfortunately, apart from skin malignancies, to date, the clinical and preclinical studies regarding skin diseases are still very few and surely need to be implemented in order to apply the knowledge and clarify the various critical and contradictory findings that characterize many skin disorders. For instance, Ma et al. suggest that Notch1 inhibition obtained by γ-secretase blockade induced by *N*-*S*-phenylglycine t-butylester (DAPT) can effectively alleviate the severity of mouse Psoriasis-like skin inflammation [143]. Nevertheless, in Psoriasis, as widely explained in the previous paragraphs, both upregulation or downregulation of Notch signaling have been reported. Therefore, we are convinced that this contribution will be significant for the integration of pathogenetic aspects and its functional consequences on the clinical phenotypes. The progressive definition of this integration is increasingly essential to create a tailored therapeutic approach.

## Figures and Tables

**Figure 1 ijms-21-04214-f001:**
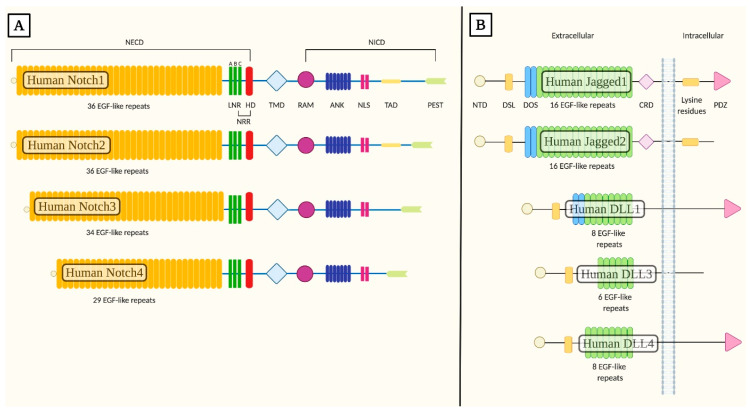
Structural organization of human Notch receptors and human canonical Notch ligands. (**A**) Mammalian cells express four different Notch receptors, Notch1–4. All receptors are single-pass type I transmembrane proteins. The notch extracellular domain (NECD) comprises 29 up to 36 epidermal growth-like factor tandem repeats (EGF-like repeats), followed by the negative regulatory region (NRR), composed by three cysteine-rich Lin12-Notch repeats (LNR-A, -B, -C) and a heterodimerization domain (HD). The single transmembrane domain (TMD) is directly followed by the Notch intracellular domain (NICD) consisting of a Recombination Signal Binding Protein for Immunoglobulin Kappa J Region (RBPjκ) association module (RAM), seven ankyrin repeats (ANK), two nuclear localization signals (NLS) and a transactivation domain (TAD) that retains conserved proline/glutamic acid/serine/threonine-rich motifs (PEST). (**B**) Five canonical Notch ligands have been described in mammals and they are generally referred to as canonical Delta/Serrate/Lag-2 (DSL) ligands. These proteins belong to the Serrate family of ligands (Jagged1 and Jagged2) and to the Delta-like family of ligands (DLL1, DLL3 and DLL4). The extracellular region possesses a conserved structural organization with an N-terminal domain (NTD), followed by a Delta/Serrate/Lag-2 (DSL) domain flanked by the Delta and OSM-11-like region (DOS) and multiple epidermal-growth-factor-like tandem repeats (EGF-like repeats). The NTD, together with the DSL, DOS and EGF-like motifs, is required for ligand binding to Notch receptors. Jagged1 and Jagged2 possess a juxtamembrane cysteine-rich domain (CRD) not present in Delta-like ligands. In the intracellular region, most Serrate ligands present numerous lysine residues involved in ligand signaling. Always within the intracellular C-terminal region, most DSL ligands express a PSD-95/Dlg/ZO-1 (PDZ) motif that is required for interactions with the cytoskeleton.

**Figure 2 ijms-21-04214-f002:**
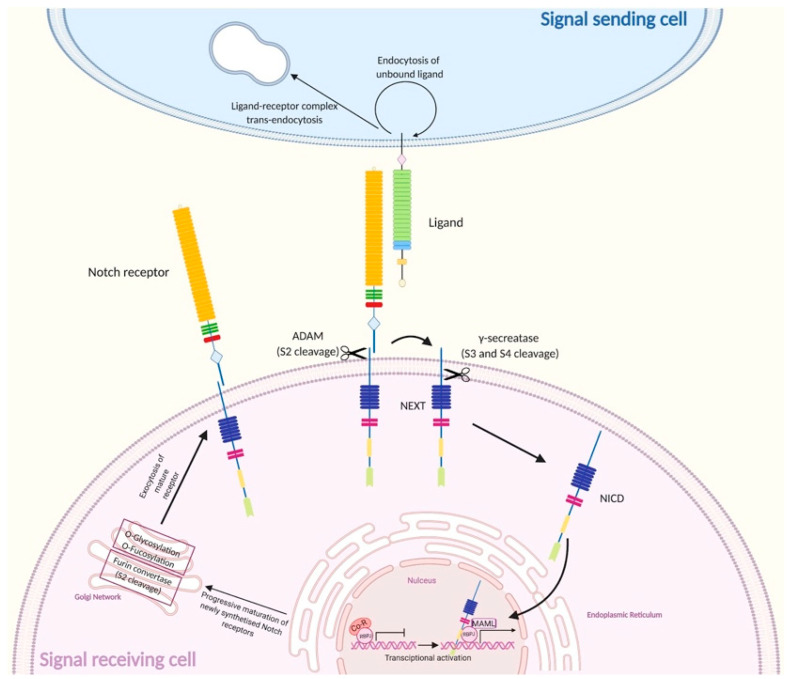
Overview of canonical Notch signaling. Notch receptors are newly translated within the endoplasmic reticulum, processed by a furin convertase (S1 cleavage) and subjected to O-linked and N-linked glycosylations in the Golgi compartment. Once on the cell surface, the receptor is activated by binding to a ligand on a juxtaposed cell. Following ligand binding, Notch signaling is initiated when the trans-endocytosis of ligand-receptor complexes in the neighboring cell induces a conformational change in the receptors, leading to the exposure of S2, site cleaved by ADAM metalloproteases (S2 cleavage) to generate a membrane-tethered partition of Notch, namely Notch extracellular truncation (NEXT). The cleavage in S2 exposes the S3 and S4, allowing the subsequent proteolytic cleavage in these sites by the γ-secretase complex (S3/S4 cleavage), ultimately leading to the release of an intracellular active fragment, the NICD. The NICD migrates in the nucleus and binds to CBF1/RBP-Jκ/Su(H)/Lag-1 (CSL), known as RBP-Jκ in vertebrates, that under basal conditions is known to function as a transcriptional repressor by associating with ubiquitous co-repressor (Co-R) proteins, and to transcriptional activators of the Mastermind-like (MAML) family, generating the ternary Notch transcription complex that initiates the transcription of downstream target genes. Membrane trafficking and endocytosis is crucial to regulate the availability of receptors and ligands on the cell surface. In the signal receiving cell, pointed arrows (→) define the orderly succession of events ranging from the synthesis and processing of Notch receptor, to its activation on the plasma membrane and subsequent induction of the transcriptional machinery. In the signal sending cell, the → the trans endocytosis of the ligand-receptor complex, while the rounded arrow (↺) describes the endocytosis of the unbound ligand.

**Figure 3 ijms-21-04214-f003:**
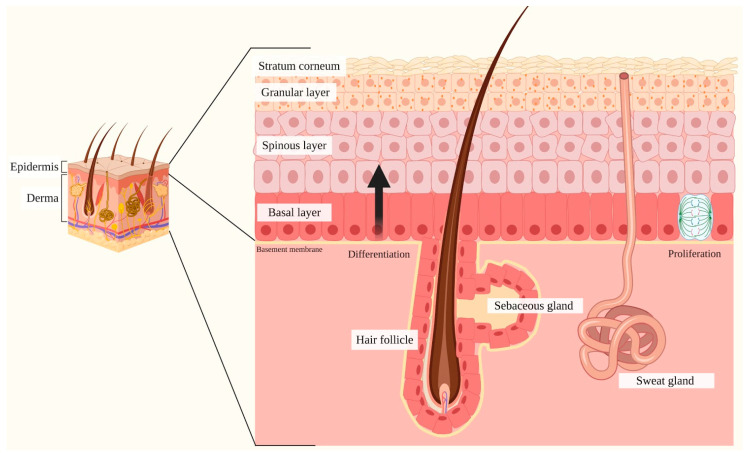
Schematic representation of the skin. The skin is composed of two layers: the epidermis and the dermis. The epidermis, the uppermost partition, is defined as a stratified squamous epithelium comprising the interfollicular epidermis (IFE) and various skin appendages (hair follicles, sebaceous glands and sweat glands). The IFE is primarily composed of progressively differentiated keratinocytes organized in specific layers, which include the basal layer (the deepest portion of the epidermis), stratum spinosum, stratum granulosum and stratum corneum (the most superficial portion of the epidermis). The basal layer comprises mitotically active cells that generate during every cell division process, either stem cells that self-renew or transient amplifying cells that gradually undergo terminal differentiation by migrating upwards towards the stratum corneum. This process requires specific signals released by the various skin appendages and contemplates the acquisition of layer-specific characteristics by keratinocytes, including the expression of epidermal keratins and transcriptional activators. Once in the stratum corneum, keratinocytes are completely keratinized, metabolically inactive and are released through desquamation. The epidermis is physically and functionally separated from the surrounding dermis by the basement membrane. The dermis is a connective tissue layer interposed between the epidermis and the subcutaneous tissue, involved in the protection and support of the skin and the deeper layers, in aiding sensation and in assisting thermoregulation.

**Figure 4 ijms-21-04214-f004:**
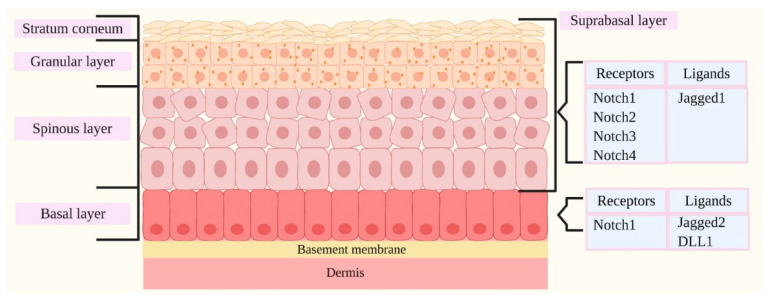
Distribution of Notch ligands and receptors in human skin. Notch1–4 receptors and their ligands are involved in the maintenance of epidermal homeostasis principally by regulating proliferation and differentiation programs within the epidermal cells, mediating the balance between proliferating basal progenitors and the terminally derived differentiated progeny, ultimately leading to the formation of the epidermal barrier. The distinctive expression patterns of Notch receptors and their ligands within the different epidermal layers seem to be associated with the activation of layer-specific target genes. Notch1–4 receptors have been reported to be expressed in the suprabasal layers, while Notch1 receptor seems to be widely expressed in the basal layer. Jagged1, Jagged2 and Delta-like 1 (DLL1) are the Notch ligands detected in the epidermis. The expression of Jagged1 has been reported to be predominant in the suprabasal layer, while Jagged2 and DLL1 have been primarily identified in the basal layer.

**Figure 5 ijms-21-04214-f005:**
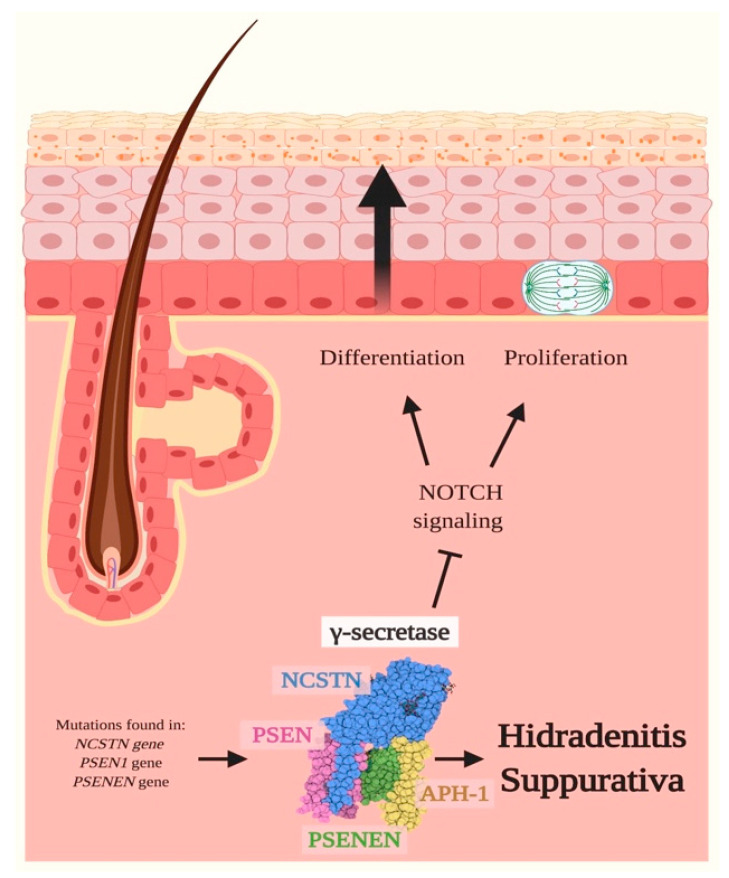
Hidradenitis Suppurativa and Notch signaling. Hidradenitis Suppurativa (HS) is a chronic inflammatory skin disease affecting the pilosebaceous unit, in which up to 35% of cases present a positive family history. These patients frequently carry mutations in *NCSTN*, *PSEN1* and *PSENEN* genes, respectively encoding for nicastrin (NCSTN), presenilin 1 (PSEN) and presenilin enhancer 2 (PSENEN) (encoding for protein). These genes encode for three of the four subunits of the γ-secretase multiprotein complex, a transmembrane protease involved in the cleavage of Notch receptors, and their haploinsufficiency results in the dysfunction of the complex. As a consequence, the γ-secretase is not able to cleave and activate Notch receptors and therefore induces an impairment of Notch signaling, ultimately resulting in augmented levels of epidermal cell proliferation and differentiation. Therefore, alterations in Notch signaling might underlie, at least partially, the initial steps of HS onset, which includes infundibular hyperkeratosis and hyperplasia of follicular epithelium that anticipate the follicular occlusion and disruption. Pointed arrows (→) define the induction of the indicated process or associated function, while the truncated arrow (Τ) designates the inhibition of the described process.

**Figure 6 ijms-21-04214-f006:**
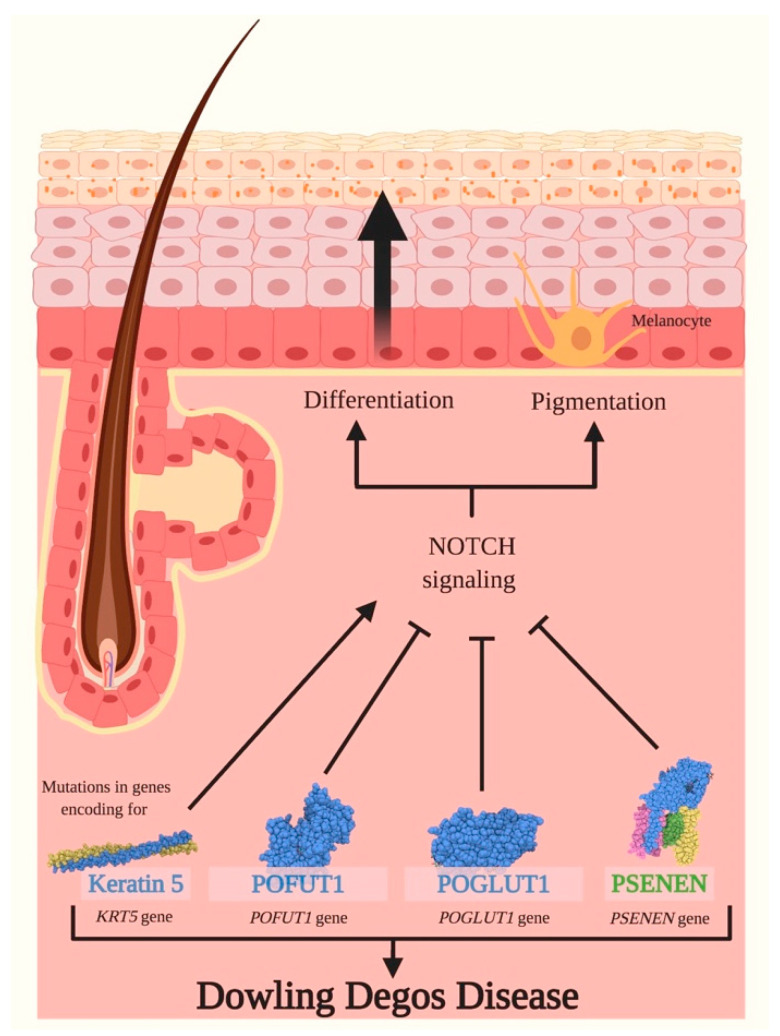
Dowling Degos Diseases and Notch signaling. Dowling Degos Disease (DDD) is a rare skin genodermatosis characterized by hyperkeratotic papules and reticulate hyperpigmentation. Classical DDD is induced by mutations in *KRT5* gene encoding for keratin 5, and the latter is a fundamental protein required for the synthesis of intermediate filaments, principally in the epidermal cells of the stratum basale. The loss of expression of *KRT5* during epidermal differentiation seems to be linked to an activation of Notch signaling, therefore negatively impacting epithelial remodeling. Other genes involved in DDD are *POFUT1*, *POGLUT1* and *PSENEN*, encoding for GDP-fucose protein O-fucosyltransferase 1 (POFUT1), Protein O-glucosyltransferase 1 (POGLUT1) and presenilin enhancer protein 2 (PSENEN), respectively. *POFUT1* mutations lead to nonsense-mediated decay of the transcript, to impaired Notch signaling and to an abnormal pigmentation of the skin. Mutations in *POGLUT1* gene presumably cause a nonsense-mediated decay of mRNA resulting in haploinsufficiency, in a blockade of the Notch pathway and in promoting cell differentiation. Pathogenic *PSENEN* variants impair Notch signaling and cause an aberrant differentiation and pigmentation of the epidermis, leading to reticulate pigmentation, comedones, follicular hyperkeratosis, nodules and scars. Pointed arrows (→) define the induction of the indicated process or associated function, while the truncated arrow (T) designates the inhibition of the described process.

**Figure 7 ijms-21-04214-f007:**
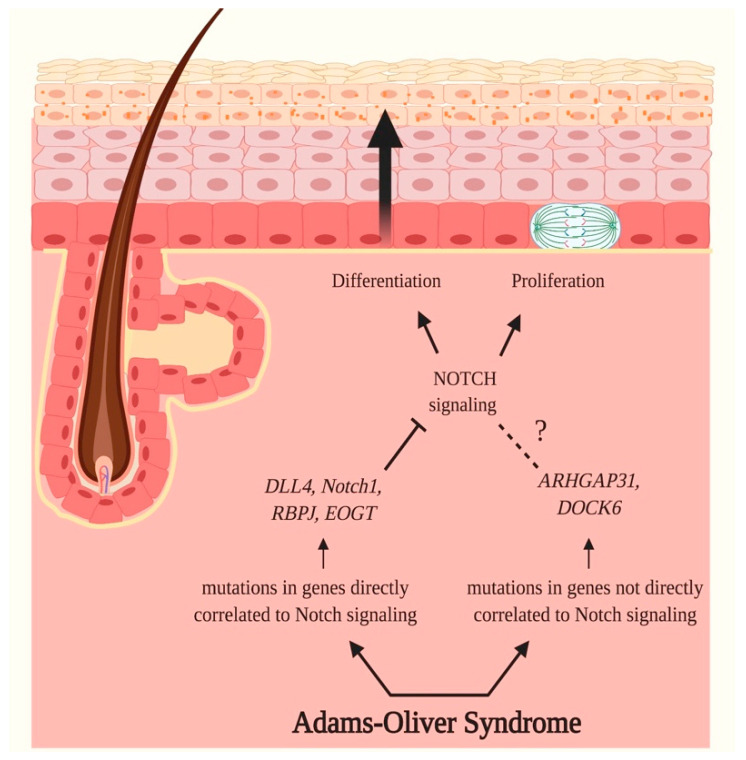
Adams–Oliver Syndrome and Notch signaling. Adams–Oliver Syndrome (AOS) is a rare inherited disorder characterized by aplasia cutis congenita of the scalp, terminal transverse limb defects, vascular anomalies, pulmonary and portal hypertension, and retinal hypervascularization. 71% of identified mutations in AOS cases have been identified in genes directly correlated to Notch signaling, therefore suggesting the crucial role of this pathway in the onset of AOS. Mutations have been registered in *Notch1*, *DLL4*, *RBP-Jκ* and *EOGT* genes, encoding, respectively, for Notch1, Delta-like 4 ligand (DLL4), Recombination Signal Binding Protein for Immunoglobulin Kappa J Region (RBP-Jκ) and EGF domain-specific O-linked N-acetylglucosamine transferase (EOGT). The identified genetic variants impact the maturation of receptors, receptor-ligand binding and interaction, and ultimately the transcription of target genes. As a consequence, these mutations cause a disruption of Notch signaling, therefore negatively impacting proliferation and differentiation of epidermal cells. The remaining 29% of AOS cases carry causative variants in *ARHGAP31*, encoding for the Rho GTPase-activating protein 31 (ARHGAP31), and *DOCK6*, encoding for guanine nucleotide exchange factor (GEF), which are not directly associated with Notch signaling. The products of these genes are regulatory proteins implied in the formation of actin cytoskeleton and are consequently involved in cell morphology, cell migration, survival and division; therefore, variants in these genes are thought to negatively regulate these fundamental processes. Pointed arrows (→) define the induction of the indicated process or associated function, while the truncated arrow (T) designates the inhibition of the described process. The dotted line (---) indicates a proposed association that still needs to be clarified.

**Figure 8 ijms-21-04214-f008:**
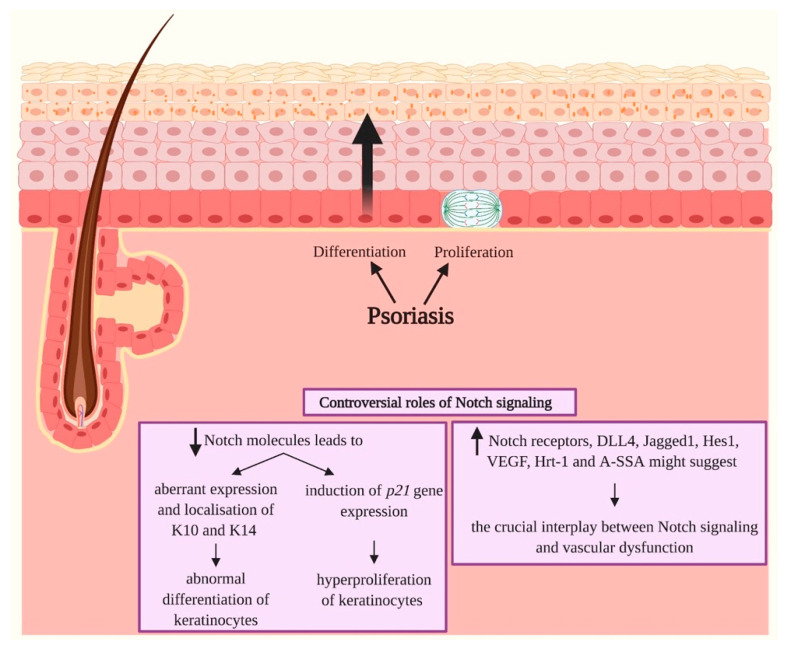
Psoriasis and Notch signaling. Psoriasis is defined as a chronic immune-mediated and inflammatory skin disease characterized by hyperproliferation and aberrant differentiation of epidermal cells, primary cutaneous manifestation and a strong genetic predisposition. The role of Notch signaling in Psoriasis seems to be controversial since both downregulations and upregulations of Notch molecules might be responsible for the onset of the skin disease. A decrement in Notch molecules results primarily in an abnormal differentiation of keratinocytes due to an aberrant expression and localization of keratin 10 (K10) and keratin 14 (K14) in the epidermis, and in the induction of *p21* gene expression leading to hyperproliferation of keratinocytes. A hyperactivation of Notch signaling has also been registered in the skin of psoriatic patients. High expression of Notch receptors, Jagged1, hairy/enhancer of split 1 (Hes1), vascular endothelial growth factor (VEGF), Ring-box protein HRT1 (Hrt-1) and Acute-phase Serum Amyloid A (A-SSA) have been registered, and they suggest the tight link between Notch signaling and vascular dysfunction in Psoriasis. Pointed arrows (→) define the induction of the indicated process or associated function. The arrows facing upwards (↑) indicate an upregulation, while the arrows facing downwards (↓) designate a downregulation of Notch molecules.

**Figure 9 ijms-21-04214-f009:**
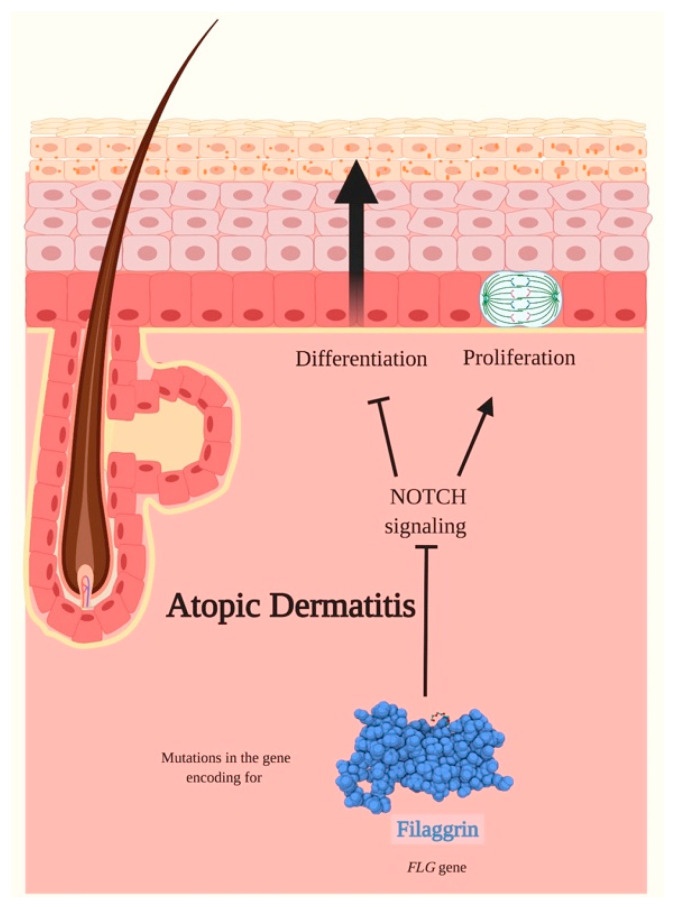
Atopic Dermatitis and Notch signaling. Atopic Dermatitis (AD) is a common, chronic and pruritic inflammatory disease. The pathogenesis of AD is multifactorial and occurs through a combination of disturbed epidermal differentiation with abnormalities in the skin barrier functions, immune dysregulation, and genetic and environmental factors. Mutations in the *FLG* gene encoding for filaggrin have been registered to be the most common variants in AD familial forms. Filaggrin is a late epidermal differentiation protein that possesses a high keratin-binding affinity, interacts exclusively with intermediate filaments to form highly insoluble keratin scaffolds necessary to guarantee the progressive differentiation of keratinocytes and ultimately form the skin barrier. Loss-of-function mutations in the *FLG* gene are common in AD and cause a loss of filaggrin expression and therefore to a disturbed epidermal differentiation. Notch signaling is required for the late-stage granular layer differentiation and correct filaggrin processing in the epidermis. A downregulation of Notch receptors expression has been identified in lesional skin of AD. The inactivation of Notch receptors seems to promote the proliferation of epidermal cells leading, amongst other manifestations, to hyperkeratosis and to impair the correct differentiation program in keratinocytes. Pointed arrows (→) define the induction of the indicated process or associated function, while the truncated arrow (T) designates the inhibition of the described process.

**Figure 10 ijms-21-04214-f010:**
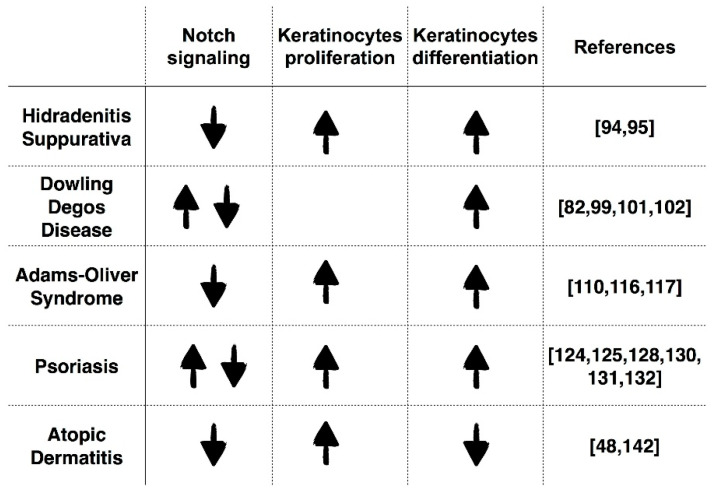
Schematic representation of the interactions between Notch signaling, proliferation and differentiation of keratinocytes in skin diseases. The role of Notch signaling, proliferation and differentiation of keratinocytes in Hidradenitis Suppurativa, Dowling Degos Disease, Adams–Oliver Syndrome, Psoriasis and Atopic Dermatitis. The arrows facing upwards (↑) indicate an upregulation, while the arrows facing downwards (↓) designate a downregulation, of Notch signaling, keratinocyte proliferation and differentiation in the various skin diseases.

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
