# Peer review of "Pleiotropic Role of Notch Signaling in Human Skin Diseases"

_ijms, 2020, doi:10.3390/ijms21124214_

Round 1
Reviewer 1 Report
This is a timely written review article regarding the role of notch signaling in human skin diseases. The authors discussed skin diseases correlated to alterations in Notch signaling including Hidradenitis Suppurativa, Dowling Degos Disease, Adams-Oliver Syndrome, Psoriasis and Atopic Dermatitis. I do have some comments and suggestions to make this review more comprehensive, which are as follows:
- It seems to me that the Figures are generated by the authors. However, the way they are mentioned in the text, it looks like they derived the figures from the referenced paper. Please clarify and change the writing in the text to make it clear. Line : 96, 139, 235, 283, 350 etc.
- Discuss more about non-canonical Notch signaling. Please see these manuscripts: PMID: 25538890, PMID: 28955033 , PMID: 30030827, PMID: 31057403, PMID: 29160307
- It’s nice to have a table showing the literatures highlighting the role of Notch in the pathogenesis of different skin diseases
- Briefly discuss the Notch therapeutics of skin diseases at least in preclinical studies (animal model).
- Need a reference for line 51
- Figure 1 and 2 can be combined into one figure
Author Response
We would like to sincerely thank the reviewers; their criticisms and valuable suggestions really helped us to render the text more readable and to ameliorate the quality of our review, which has been modified accordingly.
Please find attached the point-to-point answers to the criticisms raised by the reviewers.
Reviewer 1
Comments and Suggestions for Authors
This is a timely written review article regarding the role of notch signaling in human skin diseases. The authors discussed skin diseases correlated to alterations in Notch signaling including Hidradenitis Suppurativa, Dowling Degos Disease, Adams-Oliver Syndrome, Psoriasis and Atopic Dermatitis. I do have some comments and suggestions to make this review more comprehensive, which are as follows:
- It seems to me that the Figures are generated by the authors. However, the way they are mentioned in the text, it looks like they derived the figures from the referenced paper. Please clarify and change the writing in the text to make it clear. Line : 96, 139, 235, 283, 350 etc.
Response: all figures have been designed and draw by the authors. We generated all figures using Biorender.com as stated in the acknowledgments section. We have inverted in the text the order of appearance of the literature references and the mentioned figure. By doing so we intend to clarify that the figures are not derived from the referenced papers.
- Discuss more about non-canonical Notch signaling. Please see these manuscripts: PMID: 25538890, PMID: 28955033 , PMID: 30030827, PMID: 31057403, PMID: 29160307
Response: The text has been modified as suggested and the new references have been included in the text.
- It’s nice to have a table showing the literatures highlighting the role of Notch in the pathogenesis of different skin diseases
Response: At figure 11 the image that already shows a schematic representation of the role of Notch signaling in the skin diseases, as discussed in the review, has been implemented by adding the references.
- Briefly discuss the Notch therapeutics of skin diseases at least in preclinical studies (animal model).
Response: As suggested, we added a paragraph relative to Notch therapeutics in the conclusion section.
- Need a reference for line 51
Response: We added the reference as suggested.
- Figure 1 and 2 can be combined into one figure
Response: We combined the Figure 1 and 2 into one figure (Figure 1) as suggested.

Reviewer 2 Report
In this complete and interesting review, the authors provide an overview on the role of Notch in skin diseases. The review is well structured and with a clear focus, especially the second part on the skin diseases is precise and coherent. Figures are nice and convey the message unambiguously. The first part on the Notch pathway on the other hand, needs additional revisions in both language and style. Some concepts are redundant between the first (Notch pathway) and the second part (skin diseases) and should be homogenized. The manuscript can overall benefit from corrections from an English native speaker.
In details:
Text
Lines 46-48 need rephrasing
Lines 113-115 need rephrasing
Lines 614-619 the concepts have been largely described before, please homogenize
Line 67-68 the ligand is not adjacent to the receptor (but in the case of cis-inhibition)
Line 81 transcription can be activated or inhibited
Session 2.2 and 2.3 should have a more detailed title and should avoid redundancy
It is unclear why the title of the manuscript is written in capital letters
Figures and references
In the legend of Figure 1 “followed by” is duplicated
Some references in the text are still in the preliminary form of draft writing (the doi/PMID number in given instead a proper listing). Please update the list.
Reference number 1 should be substituted with a more appropriate one, such as:
doi:10.1126/science.271.5257.1826.
doi:10.1038/nature03589.
doi:10. 1038/nrm.2016.94
DOI: 10.1038/nrm2009
Author Response
We would like to sincerely thank the reviewers; their criticisms and valuable suggestions really helped us to render the text more readable and to ameliorate the quality of our review, which has been modified accordingly.
Please find attached the point-to-point answers to the criticisms raised by the reviewers.
Reviewr 2
Comments and Suggestions for Authors
In this complete and interesting review, the authors provide an overview on the role of Notch in skin diseases. The review is well structured and with a clear focus, especially the second part on the skin diseases is precise and coherent. Figures are nice and convey the message unambiguously. The first part on the Notch pathway on the other hand, needs additional revisions in both language and style. Some concepts are redundant between the first (Notch pathway) and the second part (skin diseases) and should be homogenized. The manuscript can overall benefit from corrections from an English native speaker.
In details:
Text
Lines 46-48 need rephrasing
Response: The text was amended as suggested.
Lines 113-115 need rephrasing
Response: The text was amended as suggested.
Lines 614-619 the concepts have been largely described before, please homogenize
Response: The text was amended as suggested.
Line 67-68 the ligand is not adjacent to the receptor (but in the case of cis-inhibition)
Response: The text was amended as suggested.
Line 81 transcription can be activated or inhibited
Response: The text was amended as suggested.
Session 2.2 and 2.3 should have a more detailed title and should avoid redundancy
Response: We modified the title of the session 2.2 and 2.3
It is unclear why the title of the manuscript is written in capital letters
Response: We modified the title letters as suggested.
Figures and references
In the legend of Figure 1 “followed by” is duplicated
Response: We modified as suggested
Some references in the text are still in the preliminary form of draft writing (the doi/PMID number in given instead a proper listing). Please update the list.
Reference number 1 should be substituted with a more appropriate one, such as:
doi:10.1126/science.271.5257.1826.
doi:10.1038/nature03589.
doi:10. 1038/nrm.2016.94
DOI: 10.1038/nrm2009
Response: We updated the references as suggested.
